# Molecular Genetic Characteristics of Different Scenarios of Xylogenesis on the Example of Two Forms of Silver Birch Differing in the Ratio of Structural Elements in the Xylem

**DOI:** 10.3390/plants10081593

**Published:** 2021-08-02

**Authors:** Natalia A. Galibina, Tatiana V. Tarelkina, Olga V. Chirva, Yulia L. Moshchenskaya, Kseniya M. Nikerova, Diana S. Ivanova, Ludmila I. Semenova, Aleksandra A. Serkova, Ludmila L. Novitskaya

**Affiliations:** Forest Research Institute, Karelian Research Centre of the Russian Academy of Sciences, 11 Pushkinskaya St., Petrozavodsk 185910, Russia; karelina.t.v@gmail.com (T.V.T.); tchirva.olga@yandex.ru (O.V.C.); tselishcheva.yulia@mail.ru (Y.L.M.); knikerova@yandex.ru (K.M.N.); dszapevalova@mail.ru (D.S.I.); mi7enova@gmail.com (L.I.S.); serkovaaleksandra1996@yandex.ru (A.A.S.); ludnovits@rambler.ru (L.L.N.)

**Keywords:** figured wood, xylem vessels, xylem fibrous tracheids, xylem parenchyma cells, *VND1*, *VND7*, *NST1*, *SND1*, *ARF5*, *HB8*

## Abstract

Silver birch (*Betula pendula* Roth) is an economically important species in Northern Europe. The current research focused on the molecular background of different xylogenesis scenarios in the birch trunks. The study objects were two forms of silver birch, silver birch trees, and Karelian birch trees; the latter form is characterized by the formation of two types of wood, non-figured (straight-grained) and figured, respectively, while it is currently not clear which factors cause this difference. We identified *VND*/*NST*/*SND* genes that regulate secondary cell wall biosynthesis in the birch genome and revealed differences in their expression in association with the formation of xylem with different ratios of structural elements. High expression levels of *BpVND7* accompanied differentiation of the type of xylem which is characteristic of the species. At the same time, the appearance of figured wood was accompanied by the low expression levels of the *VND* genes and increased levels of expression of *NST* and *SND* genes. We identified *BpARF5* as a crucial regulator of auxin-dependent vascular patterning and its direct target—*BpHB8*. A decrease in the *BpARF5* level expression in differentiating xylem was a specific characteristic of both Karelian birch with figured and non-figured wood. Decreased *BpARF5* level expression in non-figured trees accompanied by decreased *BpHB8* and *VND/NST/SND* expression levels compared to figured Karelian birch trees. According to the results obtained, we suggested silver birch forms differing in wood anatomy as valuable objects in studying the regulation of xylogenesis.

## 1. Introduction

The xylem of woody plants is characterized by a wide range of its structural elements—vessels (water conduction function), fibrous tracheids (mechanical function as the main one and water conduction function), fibers (mechanical function), and radial and axial parenchyma cells (transport function and nutrient storage function). The whole variety of xylem structural elements originates from the stem cells of the lateral meristem called cambium. The differentiation stages of cambial derivatives into xylem elements include cell enlargement, the formation of a secondary cell wall (SCW), and in the case of fibers and vessels—programmed cell death (PCD) and autolysis of the cell content. Unlike poorly differentiated parenchyma cells, fibers and vessels are dead cells that consist only of cell walls. Cell walls in woody plants account for up to 90% of the dry weight. Compared to the thin primary cell walls (PCW), SCW are much thicker and account for most cellulosic biomass that serves as a renewable resource for biofuel production [1,2]. Lately, because of the growing interest in clean bioenergy and biofuels, significant progress has been achieved in understanding the ways of regulating SCW deposition and PCD [3,4,5]. *Arabidopsis thaliana* (L.) Heynh. is a widespread experimental model for studying the molecular genetic mechanisms of regulation of xylogenesis [6]. Due to the sequencing of the genomes of tree species from both gymnosperms (*Picea abies* (L.) H. Karst [7], *Picea glauca* (Moench) Voss [8]) and angiosperms (*Populus trichocarpa* Torr. & A. Gray ex Hook [9], *Eucalyptus grandis* W. Hill ex Maiden [10], *Betula pendula* Roth [11], *Betula platyphylla* Sukaczev [12]), it is now possible to uncover the molecular mechanisms controlling the formation of both coniferous and deciduous wood.

Now *P. trichocarpa* is considered a model species for studying the molecular genetic aspects of the xylogenesis regulation in woody plants [13,14]. However, a specific for Salicaceae Mirb. genome duplication event at 60 MYA [9] led to the emergence of a large number (about 8000) duplicated gene pairs [15], and many of them had a different function compared to the homologous genes of *A. thaliana*. In addition, axial parenchyma is absent or extremely rare in the xylem of *P. trichocarpa* [16], while the proportion of this parenchyma type can reach 30% or more in the xylem of other species of woody plants of the temperate zone [17].

Meanwhile, *B. pendula*, a fast-growing woody plant, is a valuable source of woody raw materials in Northern Europe [18]. The sequencing and publication of the silver birch genome in 2017 opened up broad opportunities for the molecular genetic studies of xylogenesis using this species as an example [11]. In addition, the whole-genome sequencing of the economically valuable Asian white birch species *B. platyphylla* had been recently performed and had shown a high degree of similarity of the genomes of the two birch species [12].

The objects of the present study were two forms of silver birch differing in wood texture: silver birch (*B. pendula* var. *pendula*) which forms a typical straight-grained wood (Figure 1a,c,e); and a form of the silver birch—Karelian birch (*B. pendula* var. *carelica* (Merckl.) var. Hämet-Ahti with figured wood (Figure 1b,d,f–i). The figured wood of Karelian birch has a marble-like pattern, and its timber is one of the most expensive in Northern Europe [19]. Its ornamental properties are associated with its macro- and microstructure features: pearly luster is due to the swirl of wood structural elements and figured pattern is generated by large inclusions of parenchyma cells [20,21,22]. The more parenchyma inclusions there are, the richer the figure is (Figure 1b,g–i). While choosing the study objects, we proceeded from the assumption that any deviation from the normality can help comprehend the mechanisms of the normal process regulation deeper and more comprehensively. The structural anomalies of wood are most pronounced in Karelian birch compared to all tree birch species [23,24]. These abnormalities are characterized by a wide variety of manifestations in ontogeny and by a high level of endogenous variability; their appearance, development, and attenuation depend on the influence of environmental factors [25]. The figured wood is a hereditary trait, but even with controlled pollination, part of the trees in the progeny would still have straight-grained wood, the so-called non-figured Karelian birch trees [20,26,27]. High endogenous variability makes Karelian birch a unique object for studying mechanisms of wood formation [22,23,28].

The original Karelian birch wood is formed as a result of the cambium deviations [20,24,27,29]. The program of cell death leading to the formation of vessels and fibers of the xylem and sieve elements of the phloem does not start in the zones of development of structural abnormalities; the differentiating cambial derivatives preserve the protoplast and turn into the storage parenchyma cells, which accumulate large amounts of storage substances [20,21,22]. The formation of abnormal patterned wood is associated with suppressing the sucrose synthase pathway [30] and activating the apoplastic path [31,32] of sucrose utilization. At the same time, the cellulose content in the abnormal tissues decreases [33].

The transcriptional regulatory pathways play a pivot role in directing the cell fate of cambial stem cells and initiating the transcriptional program controlling SCW formation [34,35]. The three layers of regulators, including NAC (NO APICAL MERISTEM, ATAF1, ATAF2, and CUP-SHAPED COTYLEDON 2) domain master regulators, two MYB domain regulators, and many other regulators, are directly involved in regulating SCW biosynthetic genes [3,4,36]. The NAC proteins are critical regulators of many developmental processes [8,37,38], including SCW formation [39] and biotic and abiotic stress responses [40,41,42]. The specific transcriptional switches that regulate SCW biosynthesis and belong to the NAC family include VASCULAR-RELATED NAC-DOMAIN1 (VND1-VND7), NAC SECONDARY WALL THICKENING PROMOTING FACTOR 1 (NST1, NST2), and NST3/SECONDARY WALL-ASSOCIATED NAC DOMAIN PROTEIN 1 (SND1) [34,35,43,44,45,46,47]. We assumed that the change in the ratio of the structural elements in the mature xylem is probably accompanied by the different expression levels of the genes encoding master regulators (VND/NST/SND) in the differentiating xylem.

Plant hormone auxin has been known for many years to play a crucial role in initiating vascular tissues and xylem formation [48,49,50]. The earliest stages of embryo development involve the coordinated action of auxin synthesis, transport, and signaling [51]. Auxin perception starts with auxin binding to TIR1 (TRANSPORT INHIBITOR RESPONSE1)/AFB (AUXIN SIGNALING F-BOX) receptors. It leads to subsequent degradation of the Aux/IAA (Aux/INDOLE-3-ACETIC ACID) proteins that repress auxin signaling via physical interactions with auxin response factor (ARF) proteins [3,4,36,52,53,54]. The auxin-stimulated protein turnover of Aux/IAAs releases the transcriptional activity of their partner ARFs to activate downstream auxin-responsive gene expression [54]. Different Aux/IAA-ARF modules regulate corresponding auxin-responsive genes and developmental processes [54,55,56,57]. In *Arabidopsis*, ARF5/MP binds to the promoter of *ARABIDOPSIS THALINA HOMEOBOX 8* (*ATHB8*), which plays essential roles in the differentiation of the primary xylem, the secondary xylem, and interfascicular fibers [9,30,31]. In *Populus × tomentosa* Carrière, functional characterization of the PtoIAA9–PtoARF5 module revealed auxin-dependent differentiation of the secondary xylem derived from the cambium. It demonstrated their roles in orchestrating xylem cell specification, woody cell size, and vessel density. The *PtoHB7* and *PtoHB8*, encoding HD-ZIP III transcription factors, are direct targets of the PtoIAA9–PtoARF5 module. Moreover, it was shown that *PtoHB8* and *PtoARF5* regulated *WND6A* and *WND6B* expression, which were the closest homologs of Arabidopsis *VND6* and *VND7* [54].

Thus, this work aimed to identify possible molecular genetic differences between various scenarios of xylogenesis in silver birch, both non-figured and figured Karelian birch trees. We studied the anatomical features of mature xylem of the sampled trees; carried out identification and research of the genes encoding NAC-domain transcription factors (*VND*, *NST*, and *SND*) that regulate secondary cell wall synthesis, development, and differentiation of xylem cells; transcription factor BpHB8 that promotes xylem production from the cambial cells, as well as auxin-dependent transcription factor BpARF5. As far as we know, the function of these genes in connection with the regulation of xylogenesis of adult trees growing in natural conditions has not been previously studied.

## 2. Results

The study objects were two forms of silver birch: *B. pendula* var. *pendula*—the form of silver birch with straight-grained wood (hereafter Bp trees) and another silver birth variety *B. pendula* var. *carelica*—Karelian birch. Among Karelian birch trees, we selected trees with a high degree of the wood figure (figured *B. pendula* var. *carelica* trees, hereafter Bc FT trees) and non-figured plants that had a typical straight-grained wood with a weakly expressed texture (non-figured *B. pendula* var. *carelica* trees, hereafter Bc NF trees). The appearance of the debarked surface of the trunk and the inner surface of the bark of the studied plants are shown in Figure 2.

### 2.1. Features of the Structure of Mature Xylem in Different Forms of Silver Birch

The xylem of Bp and Bc NF trees has an ordered structure with no signs of anomalies (Figure 2a,b and Figure 3a,b). The xylem of Bc FT trees is characterized by a high level of heterogeneity [21]: abnormalities can closely coexist with structurally normal areas (Figure 2c–f and Figure 3c). Microscopic analysis of tissues showed that xylem samples of the studied trees differ from each other. The main features of the anomalous structure of the xylem of Bc FT trees are the growth-ring curves and changes in the composition of structural elements (Figure 3c).

In the mature xylem of Bp trees, the ratio of fibrous tracheids: vessels: parenchyma cells was 74:18:8 and 73:18:9 in 2019 and 2020, respectively (Figure 4a–c,e–g). In the abnormal xylem of Bc FT trees, the proportion of vessels was two times lower (Figure 4a,e), and the ratio of parenchyma cells was 1.3 times higher (Figure 4c,g) than in Bp trees. Although the proportion of fibrous tracheids in abnormal xylem did not differ from that of the xylem of Bp trees, the cell wall thickness of fibrous tracheids was increased (Figure 4d,h). We found that the vessel density was 1.7–1.9 times lower in the anomalous zones than in the wood without anomalies (Figure 5a,b). In some sections, vessels in the area of abnormalities are absent (Figure 3c).

In structurally normal xylem of Bc NF trees, the ratio of structural elements did not differ from that of Bp trees (Figure 4a–c,e–g). At the same time, the cell wall thickness of fibrous tracheids was higher than that of Bp trees but lower than that of Bc FT trees (Figure 4d,h). The number of vessels per unit area was the highest among the samples studied (Figure 5a,b).

### 2.2. NAC Family Genes Identification in the Silver Birch Genome

Previously, 108 NAC family genes have been identified in the silver birch genome, and seven genes have been reported to be homologous to *A. thaliana* Class IIB NACs [42]. We carried out a detailed analysis of these seven genes and found that 6 of them encode proteins homologous to VND, NST, and SMB/BRN of *A. thaliana* and *P. trichocarpa*. In contrast, one gene encodes a truncated protein lacking C-terminus. In the genome of silver birch, 2 VND genes and two genes belonging to the NST family were identified (Figure 6). Another two genes belonged to the SMB/BRN family. Because these genes in woody plants are expressed in root tissues [58], we did not further study the genes of the SMB/BRN family.

Birch genes encoding homologs of *A. thaliana* and *P. trichocarpa* VND1 and VND7 were located on chromosomes 4 and 7 and contained 3 and 2 introns, respectively (Table 1, Figure 7). We also identified genes encoding the homologs of *A. thaliana* and *P. trichocarpa* NST1 and NST3/SND1 (hereafter SND1), located on chromosomes 5 and 14, respectively, and contained two introns. The corresponding proteins contained subdomains A, B, C, D, E in the NAC domain located in the N-terminus and transcriptional activation regions in C-terminus sequences. They shared 48.9–57.3% and 69.2–73.5% of identical amino acids with the VNDs and NSTs of *A. thaliana* and *P. trichocarpa*, respectively.

### 2.3. Expression of the Genes Encoding NAC-Domain Transcription Factors in the Differentiating Xylem of Different Forms of Silver Birch

We assumed that the xylem formation with different anatomical characteristics in two forms of silver birch would be accompanied by changes in the expression level of genes encoding NAC-domain master regulators. The results showed that in the differentiating xylem, the number of transcripts of the genes *BpVND1*, *BpVND7*, *BpNST1*, *BpSND1* differs in Bp, Bc NF, Bc FT trees, respectively.

In the differentiating xylem of Bp trees, the number of *BpVND7* transcripts, which was the closest homolog of *AtVND7*, was the highest, and the expression levels of *BpVND1*, *BpNST1*, and *BpSND1* genes were 2-, 13-, and 260-fold lower, respectively (Figure 8).

The expression level of *BpVND7* and *BpVND1* genes in Bc FT tress was significantly 15–17-fold lower than in Bp trees. At the same time, the number of *BpNST1* and *BpSND1* transcripts in figured plants was 2-fold and 10-fold higher, respectively (Figure 8).

Exciting data were obtained for non-figured Karelian birch plants. The expression level of *BpVND7* and *BpVND1* genes was 2-fold higher than in Bp trees, *BpNST1*–10-fold higher, and *BpSND1*–158-fold higher, respectively (Figure 8).

### 2.4. ARF Family Genes Identification in the Silver Birch Genome

In the genome of silver birch, 17 genes encoding proteins homologous to the ARF proteins of *A. thaliana* and *P. trichocarpa* were identified (Figure 9). The silver birch *ARF* family genes were located on 13 chromosomes and one contig and contained 1 to 15 introns (Table 2, Figure 10). All birch ARF proteins had a conserve N-terminal B3-like DNA-binding domain (DBD) and a conserved C-terminal dimerization (CTD) domain and shared 47.2–73.0% and 48.8–80.8% of identical amino acids with the ARFs of *A. thaliana* and *P. trichocarpa*, respectively.

### 2.5. HD-ZIP III Genes Identification in the Silver Birch Genome

Four genes encoding the HD ZIP III homologs of *A. thaliana* and *P. trichocarpa* were identified in the genome of silver birch (Figure 11). The *HD ZIP III* genes were located on four chromosomes and contained 17 introns (Table 3, Figure 12). The corresponding proteins had a structure typical of HD ZIP III and contained homeodomain, START domain, and the unique MEKHLA domain, and shared 80.7–88.3% and 86.3–92.7% of identical amino acids with the HD ZIPs III of *A. thaliana* and *P. trichocarpa*, respectively.

### 2.6. Expression of the Genes Encoding Auxin-Dependent Transcription Factor BpARF5 and Transcription Factor BpHB8 in the Cambial Zone and Differentiating Xylem of Different Forms of Silver Birch

It was previously shown that *PtoHB8* was one of the critical regulators of vascular cambium differentiation to the secondary xylem in poplar and downstream targets of PtoARF5 during wood formation [54]. Therefore, we investigated the expression level of *BpHD8* and *BpARF5* genes in the cambial zone and the differentiating xylem in two forms of silver birch.

We found that the number of *BpHB8* transcripts in Bp and Bc NF trees did not differ, both in the cambial zone and in the differentiating xylem. At the same time, in Bc FT trees, *BpHB8* expression was significantly lower than in plants with typical wood structures (Figure 13c,d).

We showed that the number of *BpARF5* transcripts from differentiating xylem from the tissue layer, including the cambial zone and the vascular phloem, was higher in Bp and Bc FT trees. In the xylem, the level of its expression in Bp trees was significantly higher than that in Karelian birch plants with figured and non-figured wood. *BpARF5* expression in Bp and Bc NF trees did not differ, while it was lower in the phloem in Bc FT trees (Figure 13a,b).

### 2.7. AuxRE cis-Elements in Promoters of Studied B. pendula Genes

To get a complete picture of the regulation of figured wood formation, we analyzed the promoter regions of the studied genes. We revealed the presence of different known and putative auxin-responsive elements (AuxREs) [59] (Table 4).

## 3. Discussion

### 3.1. Distribution of BpARF5, BpHB8, BpVND, and BpNST Genes Expression during the Formation of Straight-Grained Wood of Silver Birch

The reactivation of cambium in birch under the conditions of Karelia (Petrozavodsk) occurs in mid-late May, depending on weather conditions [29,60,61]. The cambial growth stages include differentiation of conducting phloem (~mid-May); the beginning of the xylem differentiation and the active formation of early thin-walled wood (late May–mid-June); active formation of the secondary cell walls (late June–early July). Microscopic studies showed that the plants sampled in 2019 (25 June 2019) and 2020 (26 June 2020) did not differ significantly in the stage of cambial growth (Appendix A). Active cambial divisions and subsequent differentiation of xylem cells took place in 14–15-year-old birch trees at the sampling dates.

The study of the birch cambium ultrastructure [29] showed that during the period of intensive wood growth, xylem cells with a pronounced secondary thickening of the walls could be observed at a distance of 8–10 cells from the cambium. The cell wall formation was the leading process, to which the entire metabolism of the differentiating cells was subordinated from the moment of division to forming a mature element [29]. We found that genes encoding VND proteins (*BpVND1* and *BpVND7*) were mainly expressed in the differentiating Bp trees xylem, while the number of *BpNST1* and especially *BpSND1* transcripts were deficient.

Recent studies have shown that target genes of VND and NST/SND differ in *Arabidopsis thaliana*. In xylem vessels, VND proteins regulate both PCD and SCW biosynthesis [36]. VND6 and VND7 activated the expression of a broad range of genes involved in PCD, such as xylem-specific papain-like cysteine peptidase [34,62,63,64]. Other VND family members, i.e., VND1 to VND5, duplicated functions with VND6 and VND7 in vessel development [46]. In *A. thaliana*, the NST1, NST2, and SND1 proteins regulated all three components, i.e., cellulose, hemicellulose, and lignin biosynthesis in xylem fibers [35,43,45,65]. However, the expression specificity depending on the gene group as found in *Arabidopsis* was not detected in the other plant species. In poplar, rice, and maize, the genes of both VND and NST groups were expressed in vessels and fibers [66,67,68].

The co-expression network comprises *Populus VND/NST/SND* genes, PCW and SCW *PtrCESA* genes, as well as xyloglucan and pectin biosynthetic genes associated with the primary wall formation, demonstrated three distinct subclusters with member genes closely co-varying [55]. *PtrSND1*, *PtrNST1*, and *PtrVND3-1* (homologous to *BpSND1*, *BpNST1*, and *BpVND1*, respectively) formed a subcluster connected to SCW *PtrCESA*s. According to this fact, one can conclude that *BpSND1*, *BpNST1*, and *BpVND1* regulate cellulose of SCW biosynthesis. In poplar, these subclusters also included *PtrVND7-1* (homologous to birch *BpVND7*). In our studies, the level of *BpVND7* expression in the differentiating xylem was the highest in comparison with other genes. *PttVND7-1* was strongly induced in the zone of cell maturation/PCD. At the same time, the analysis of the co-expression network revealed a positive correlation of *PtrVND7-1* neither with SCW *PtrCESAs*, nor with PCW *PtrCESAs*, nor with other cell wall biosynthetic genes in poplar. There was identified only a negative correlation with the pectin biosynthetic gene (*Potri.016G001700*) [55]. On the contrary, previously, S. Chen with colleagues [42] showed that in the xylem of *B. pendula* plants, the expression level of *BpNAC072* (*BpSND1*) and *BpNAC002* (*BpNST1*), compared to *BpNAC005* (*BpVND1*) and *BpNAC057* (*BpVND7*), was higher. It should be noted that the study was carried out on the xylem tissues of two-year-old trees, which were taken from the upper part of the stem (about 10th nodes), mainly consisted of the mature structural elements; probably, it was the reason for the differences obtained. We found that a high level of *BpVND7* expression was confined to the differentiating xylem stage in 14-year-old birch plants, including xylem cells enlargement, secondary cell wall formation, and PCD of vessels and fibrous tracheids.

We found that *BpARF5* expression was 5-fold higher in the differentiating xylem of Bp trees than in the cambial zone and the conducting phloem. Previous studies had shown that auxin signaling levels increased in the differentiating cambium derivatives, while a moderate level of signaling in the cambial stem cells was essential for cambium activity [69]. At the same time, the ARF5 restricts the number of stem cells by directly attenuating the action of the stem cell-promoting *WOX4* gene [69]. We found that the number of *BpHB8* transcripts was also higher in the differentiating xylem than in the cambial zone. This was consistent with the result that was shown for *P. tomentosa*, where PtoARF5 regulated the expression of *PtoHB7* and *PtoHB8* via binding to the auxin response elements (AuxREs) within their promoters [54].

In silico analysis of the regulatory *cis*-elements revealed that a series of known canonical (TGTCTC and TGTCCC) and putative (TGTCCC and GTCCCC) AuxREs (potential DNA binding sites for ARFs) were predicted in the 2-kb of *BpSND1*, *BpNST1*, *BpVND7*, and *BpVND1* promoters. The amount of AuxRE was the highest in the promoter of *BpVND7*. At the same time, in *Populus* genes, the auxin-responsive element (AuxRE, TGTCTC) was discovered only in the promoters of *PtrSND1*, *PtrNST1*, and *PtrVND3-1* (homologous to *BpSND1*, *BpNST1*, and *BpVND1*, respectively) [55]. Together, our data and literature data allow us to suggest the active participation of *BpVND7* in the initial stages of the xylem differentiation in Bp trees.

### 3.2. The Formation of the Auxin-Deficient Phenotype of Karelian Birch Occurs against the Background of a Decrease in the Expression of BpARF5 and BpHB8, and It Is Accompanied by a Change in the Expression of the NAC Family Genes

We showed that areas with structural abnormalities in figured Karelian birch wood contained much fewer vessels than straight-grained wood of silver birch. It was believed that free (physiologically active) auxin was the only signal required for the differentiation of vessels [49,70,71,72,73]. A reduced density of vessels in figured wood, as well as a disturbance of their spatial orientation [28], indicated a decrease in free (physiologically active) auxin [49,74]. The fact that the vascular tissues of the trunk of figured Karelian birch trees had features of the auxin-deficient phenotype had been repeatedly discussed in the literature earlier. Thus, the biochemical analysis carried out by S.V. Shchetinkin [75] showed that in the figured sections of the trunk of Karelian birch trees, the content of free (physiologically active) auxin was reduced both in comparison with non-figured areas of the same trunk and in contrast, with the silver birch trunks. Analysis of the expression of genes involved in auxin homeostasis (families *Yucca*, *PIN*, *GH3*, *UGT*) showed that the reason for the formation of abnormal wood structure in Karelian birch could be the inactivation of auxin in its conjugation reactions with amino acids and sugars [28,76].

We found that the level of *BpARF5* expression in abnormal xylem of figured Karelian birch trees was significantly reduced, from the side of the cambial zone, especially in differentiating xylem straight-grained wood of silver birch. Previous studies had shown that in places of anomalies in Karelian birch, ray initials undergo mainly anticlinal and transverse divisions instead of ordered periclinal divisions, which were dominant during the formation of typical straight-grained wood [29]. Based on the fact that the *PtoHB8*, encoding HD-ZIP III transcription factor, is one of the direct targets of the PtoARF5 [54,69], it seems pretty logical that *BpHB8* has a lower level of expression in figured trees. A decreased transcription of genes that control early xylem development was reflected in *Betula VND*/*NST*/*SND* genes expression. Notably, *BpVND7* and *BpVND1* were lower, while the expression of *BpSND1* and *BpNST1* was higher compared to straight-grained wood of silver birch.

To explain the results obtained, we used the data reported on transgenic poplar lines overexpressing stabilized *PtoIAA9* (*PtoIAA9m-OE*) [54]. PtoIAA9 contains intact Domains III and IV that mediate physical interaction with ARFs, to repress downstream auxin-responsive gene expression, which may be similar to downstream *ARF5* expression. In *P. tomentosa*, the overexpression of *PtoIAA9m* inhibited the periclinal division of cambium, thus leading to reduced wood formation. At the same time, the expression of both *PtoHB7* and *PtoHB8* were significantly decreased by 71% and 74% in the *PtoIAA9m*-OE lines [54]. We had shown that in Bc FT trees, against the background of a decrease in the xylem growth, an increase in the thickness of the cell wall of fibrous tracheids occurred. Similar changes were found in the *PtoIAA9m*-OE plants compared to WT [54]. Active cambium periclinal cell divisions lead to rapid xylem mother cell accumulation. In contrast, a low rate of cambium periclinal cell division slowly produces xylem mother cells, resulting in the absence of the early developing xylem cell layers in the *PtoIAA9m*-OE lines [54]. Collectively, our data allow us to propose that (1) a decrease in the expression level of *BpARF5* in abnormal regions could be the reason for the formation of the auxin-deficient phenotype of Karelian birch and (2) an increase in the expression level of *BpSND1* and *BpNST1* may be associated with their participation in the regulation of SCW biosynthetic genes.

### 3.3. Non-Figured Karelian Birch Plants Are of Particular Interest for the Study of the Development of Secondary Conducting Tissues

We showed that in Bc NF trees, the expression of *BpARF5* in the differentiating xylem was approximately 5-fold lower than Bp trees and did not significantly differ from that in the abnormal areas of Karelian birch. At the same time, from the side of the cambial zone, the number of *BpARF5* transcripts in Bc NF trees was even a bit higher than in Bp trees. Our data indicated a well-pronounced expression of *BpHB8* from the cambial zone and the zone of the xylem differentiation in Bc NF trees. The number of transcripts of these genes practically did not differ from that of Bp trees. Importantly, the expression of the studied *Betula VND/NST/SND* genes, especially *BpNST1* and *BpSND1*, significantly exceeded that of Bp trees. Compared to the straight-grained wood of silver birch, the fibrous tracheids’ vessel density and cell wall thickness in straight-grained wood of Bc NF trees were significantly higher. Therefore, our data suggested that (1) a decreased level of *BpARF5* expression in differentiating xylem was characteristic of Karelian birch plants, both with figured and non-figured wood; (2) at the same time, the conductive tissues of the Bc NF tree trunk, in contrast to Bc FT trees, did not exhibit the features of the auxin-deficient phenotype.

The expression of *ARF* genes is tissue-specific, varies slightly in response to hormonal stimuli, and is regulated mainly on the posttranscriptional level by microRNAs (miRNA) and trans-acting-small interfering RNAs (ta-siRNA) [77,78]. It is known that diverse auxin-triggered developmental phenotypes depend on the expression of specific auxin-responsive genes targeted by Aux/IAA-ARF pairs [54,55,56,79]. Under conditions when the auxin concentration is low, Aux/IAAs are more stable and interact with ARFs to inactivate them. At increased auxin concentration, auxin stabilizes the interaction between TIR1 and Aux/IAA proteins. It leads to the Aux/IAA degradation, resulting in ARF derepression and activating expression of specific gene sets [3,4,36,52,53,54,79].

Auxin biosynthesis and intercellular transport, auxin conjugation, and degradation processes determine auxin levels in individual cells and thus auxin distribution patterns within tissues [79,80,81,82]. We have previously shown that the auxin-deficient phenotype in Karelian birch trunks tissues is formed against the background of overexpression of genes involved in auxin biosynthesis (*Yucca*), polar auxin transport (*PIN*), and conjugation of auxin with amino acids (*GH3*) and UDP-glucose (*UGT84B1*) [28,76]. We have suggested that overexpression of *GH3* and *UGT84B1* genes encoding enzymes that participated in auxin conjugation appears to significantly affect figured wood formation in birch than overexpression of *Yucca* genes [76]. Interestingly, the expression of *UGT84B1* and *GH3* in trunk tissues of Bc NF trees was much lower than in Bc FT trees, and it had the same level in Bp trees. Moreover, Bc NF trees had a higher expression of *BpYucca3*, *BpYucca10*, and *BpYucca12* in leaves and expression of *BpYucca’s* in the cambial zone compared to Bp trees [76]. Collectively, the data obtained through this study and previously [28,76] allow us to propose that increased auxin concentration can rescue the formation of straight-grained wood in Bc NF trees against the background of low *BpARF5* expression.

## 4. Materials and Methods

### 4.1. Study Objects

The study objects were two forms of silver birch: *Betula pendula* Roth var. *pendula*—the form of silver birch with straight-grained wood and *B. pendula* var. *carelica*—Karelian birch. All plants grew in the same soil and climatic conditions on the experimental plot of the KarRC RAS near the Petrozavodsk city. The trees were 14–15 years old. Karelian birch plants were grown from seeds obtained from the controlled pollination of Karelian birch plus trees. The number of trees in each group was 5–6. Samples were collected during active cambial growth (25 June 2019, and 26 June 2020).

### 4.2. Plant Sampling

The trunk tissue samples were taken at breast height (1.3 m above ground level) on the same side (the south side). The trunk sections with the highest degree of manifestation of structural anomalies were selected for the sampling of plant material from the Karelian birch trunks. For microscopic analysis, blocks including the last 2–3 annual increments of wood were cut out (5 × 5 × 3 mm, length × width × height). For chemical analysis, «windows» of 6 × 8 cm were cut out on the trunk, and the bark was separated from the wood. During active cambial growth, the bark moves away from the wood along the expanding xylem zone. Tissue complexes (hereafter phloem), including the cambial zone (phloem mother cells, cambial initial, xylem mother cells), differentiating phloem, and mature phloem, were prepared the inner surface of the bark. The layers of differentiating xylem (hereafter xylem) were scraped off the exposed wood surface with a blade. The sampling of stem tissues was monitored under a light microscope (Appendix A). The material for the genetic analysis was frozen in liquid nitrogen and stored at −80 °C.

### 4.3. Microscopy

Three samples of the mature xylem tissue from each tree were analyzed. Sample preparation for microscopy was done as described previously [28]. Briefly, the samples were fixed in glutaraldehyde, dehydrated in a series of alcohols of rising concentrations, and embedded in Araldite-Embed-812 mixture following a well-known technique [83]. Panoramic transverse sections 2 µm thick were cut with an LKB Ultrotome IV (Sweden) and stained with Safranin 1% aqueous solution. Microscopic analysis was carried out under AxioImager A1 light microscope (Karl Zeiss, White Plains, NY, USA) equipped with an ADF PRO03 camera. Images were processed with ADF Image Capture software (ADF Optics, Ningbo, China). Anatomical measurements were made following available guidelines using panoramic xylem cross-sections with an area of 7–10 mm^2^ [84,85,86]. The proportion of various cell types in xylem was determined using the grid method [87]. Briefly, a grid of points 50 μm apart in horizontal and vertical directions was overlaid over the image of the xylem section. A minimum of 300 points for each sample was analyzed using ImageJ software (NIH, Bethesda, MD, USA). The percentage of each cell type was estimated as several points of a given type divided by the total number of analyzed points.

### 4.4. Gene Retrieval from the Silver Birch Genome by Bioinformatics Methods

The search for *NAC*, *ARF*, and *HD-ZIP III* genes was carried out using the published genome of *Betula pendula* Roth [11]. To this end, the CDS of *A. thaliana* and *P. trichocarpa NAC*, *ARF* and *HD-ZIP III* genes and the amino acid sequences of corresponding proteins were obtained from The Arabidopsis Information Resource (TAIR) database (release 13, https://www.arabidopsis.org, accessed on 1 June 2021) and Phytozome database (http://www.phytozome.net/poplar, release v3.0, accessed on 1 June 2021), respectively. The resulting sequences were then used as a BLAST search query across the genome of *B. pendula* var. *pendula* (release 1.2, https://genomevolution.org/coge, accessed on 1 June 2021) to identify homologous sequences.

The structures of candidate proteins were predicted using the National Centre for Biotechnology Information (NCBI) resource (http://www.ncbi.nlm.nih.gov/Structure/cdd/cdd.shtml, accessed on 1 June 2021) [88]. Phylogenetic analysis was carried out using MEGA X software [89]. Multiple sequence alignment for the protein sequences was performed using ClustalW. Default parameters were used to determine the best-fitting evolutionary models by the model test. Phylogenetic trees were constructed using the Maximum Likelihood method based on the Jones–Taylor–Thornton (JTT) model with 1000 bootstrap replicates [90,91,92,93]. The percent identity of *B. pendula* and *A. thaliana* proteins was determined using the EMBOSS Needle online tool (https://www.ebi.ac.uk/Tools/psa/emboss_needle/, accessed on 1 June 2021). The gene structures were analyzed using the Gene Structure Display Server (http://gsds.cbi.pku.edu.cn/, accessed on 1 June 2021) [87].

### 4.5. qRT-PCR

Isolation of total RNA was performed using an extraction STAB buffer (pH 4.8–5.0): 100 mM Tris—HCl (pH 8.0), 25 mM EDTA, 2 M NaCl, 2% STAB, 2% PVP; 2% mercaptoethanol was added to the mixture before use. The plant tissue lysates were additionally treated with DNase and RNase inhibitors (Syntol, Russia). Separation of the aqueous and organic phases was done using a mixture of chloroform-isoamyl alcohol (24:1). RNA was precipitated using 25 mM LiCl, then re-precipitation was carried out using an extraction SDS buffer: 1 M NaCl, 0.5% SDS, 10 mM Tris—HCl (pH 8.0), 1 mM EDTA [94]. RNA was re-precipitated with absolute isopropanol. The quality and quantity of the isolated RNA and synthesized cDNA were checked spectrophotometrically and by electrophoresis in 1% agarose gel.

Specific primers (Sintol, Russia) for amplification of the studied genes were designed using the software Beacon Designer 8.21 (PREMIER Biosoft) (Table 5). As a reference gene for normalization of quantitative PCR data, we assume to use *Ef1a* and *GAPDH* family [95]. Our previous research evaluated the suitability of five genes—*GAPDH1*, *Actin1*, *Ef1a (1)*, *Ef1a (2)*, and *18SrRNA*—in two forms of silver birch for use as a reference when staging qRT-PCR based on the stability of their expression. The results of studying the expression level of potential reference genes were analyzed using the NormFinder and BestKeeper programs. We showed that *Actin1*, *GAPDH1*, and *Ef1a (1)* are the most stable expressed in *B. pendula* samples (in leaves and trunk tissues) among the studied genes. In contrast, *GAPDH1* and *Ef1a (1)* genes were suggested by NormFinder as the best combination of two reference genes and had the lowest stability index [95]. Families of genes coding for (1) auxin-dependent transcription factor *BpARF5*, (2) transcription factor *BpHB8* that promotes xylem production from cambial cells, (3) NAC-domain transcription factors (*BpVND1*, *BpVND7*, *BpNST1*, *BpSND1*) that regulate secondary cell wall synthesis, development, and differentiation of xylem cells, were studied.

Amplification of samples was performed using an iCycler instrument with an iQ5 optical module (Bio-Rad) and the amplification kit with an intercalating dye SYBR Green (Evrogen). The specificity of amplification products was checked by melting the PCR fragments. PCR determined the effectiveness of PCR for primers of reference and target genes with successive dilutions of the investigated cDNA sample [96]. The relative quantity of gene transcripts (RQ) was calculated from the formula:RQ = E^−ΔCt^,(1)
ΔCt = Ct(target gene) − Ct(reference gene)(2)
where Ct was the threshold cycle values for the target and reference genes, E—effectiveness of PCR. Ct (reference gene) was calculated as the average meaning between Ct (*GAPDH1*) and Ct (*Ef1a (1)*). To determine the efficiency (E), PCR amplification was performed with each pair of primers in a series of 10-fold dilutions (10^−1^, 10^−2^, 10^−3^, 10^−4,^ and 10^−5^) of cDNA. Using Excel software, a plot of Ct versus Lg (conc. cDNA) was plotted and using the values of the slope of the curve (slope, k), the efficiency was calculated using the formula [97] (Appendix A):E = 10^(−1/k)^(3)

The level of expression of specific genes was expressed in relative units (arbitrary units).

We took three trees on 25 June 2019, to study *BpVND1*, *BpVND7*, *BpNST1*, *BpSND1* in Bp, NF, and FT trees Karelian birch, and two trunk sections were taken from each trunk.

We took three trees on 26 June 2020, for studying *BpARF5*, *BpHB8* in Bp, NF, and FT trees of Karelian birch, and three trunk sections were taken from each trunk.

### 4.6. Analysis of Promoters

2-Kbp upstream promoter sequences from the start codon (ATG) of genes were used to analyze *cis*-acting elements. The search for the known and putative AuxREs located in the promoters of the studied genes was performed using the PlantPAN 3.0 resource (http://plantpan.itps.ncku.edu.tw/, accessed on 1 June 2021) [98].

### 4.7. Statistical Data Processing

The results were statistically processed with PAST (version 4.0). Before starting the statistical analysis, raw data was initially tested for normality using the Shapiro-Wilk test. The significance of differences between variants was estimated by Mann–Whitney U-test. The significant difference was evaluated at the level of *p* < 0.05. Also, neighbor-joining clustering was used to select tree groups based on the expression of the studied genes.

All data in the diagrams appear as mean ± SD, where SD is the standard deviation. Different letters indicate significant difference at *p*-value < 0.05 according to results of Mann–Whitney U-test.

The research was carried out using the equipment of the Core Facility of the Karelian Research Centre of the Russian Academy of Sciences.

## 5. Conclusions

We have studied for the first time the molecular aspects of the differentiation of the cambial derivatives into xylem elements in different forms of silver birch, which are very different in wood anatomy. We identified *VND/NST/SND* genes that regulated SCW biosynthesis, development, and differentiation of xylem cells in the genome of silver birch and showed changes in their expression in the differentiating xylem in connection with the different ratio of fibrous tracheids: vessels: parenchyma cells in the mature xylem. We found a high expression level of *BpVND7* in Bp trees, which xylem displayed a typical for the species ratio of structural elements. The formation of the xylem in Bc FT trees (which were characterized by the reduced vessels density, the increased proportion of parenchyma, and the thicker cell walls of fibrous tracheids) was associated with reduced expression of *VND*s and increased expression of *NST/SND* genes. The formation of the straight-grained wood of Bc NF trees, which were characterized by a high density of vessels, occurred against the background of the increased expression level of all *VND/NST/SND* genes.

We identified genes of *ARF* and *HD-ZIP III* families in the genome of silver birch. Also, we investigated the expression level of *BpARF5* (a key regulator of auxin-dependent vascular patterning) and its direct target—*BpHB8* in trunk tissues of different forms of silver birch. We found that the level of *BpARF5* expression in the differentiating xylem of Karelian birch plants with both figured and non-figured wood was lower than in Bp trees. At the same time, the formation of figured wood of Bc FT trees was associated with low expression of *BpHB8*, whereas in Bc NF trees with a straight-grained wood expression of *BpHB8* was indistinguishable from that of Bp trees. Our data have suggested that the increased auxin concentration can rescue the formation of straight-grained wood in Bc NF trees with low *BpARF5* expression. Further research should focus on a detailed analysis of *Aux/IAAs*, *ARFs*, *HD ZIPs III*, and *VND/NST/SND* genes in the cambial zone and the differentiating xylem in different forms of silver birch, which will make it possible to investigate this hypothesis in the future.

## Figures and Tables

**Figure 1 plants-10-01593-f001:**
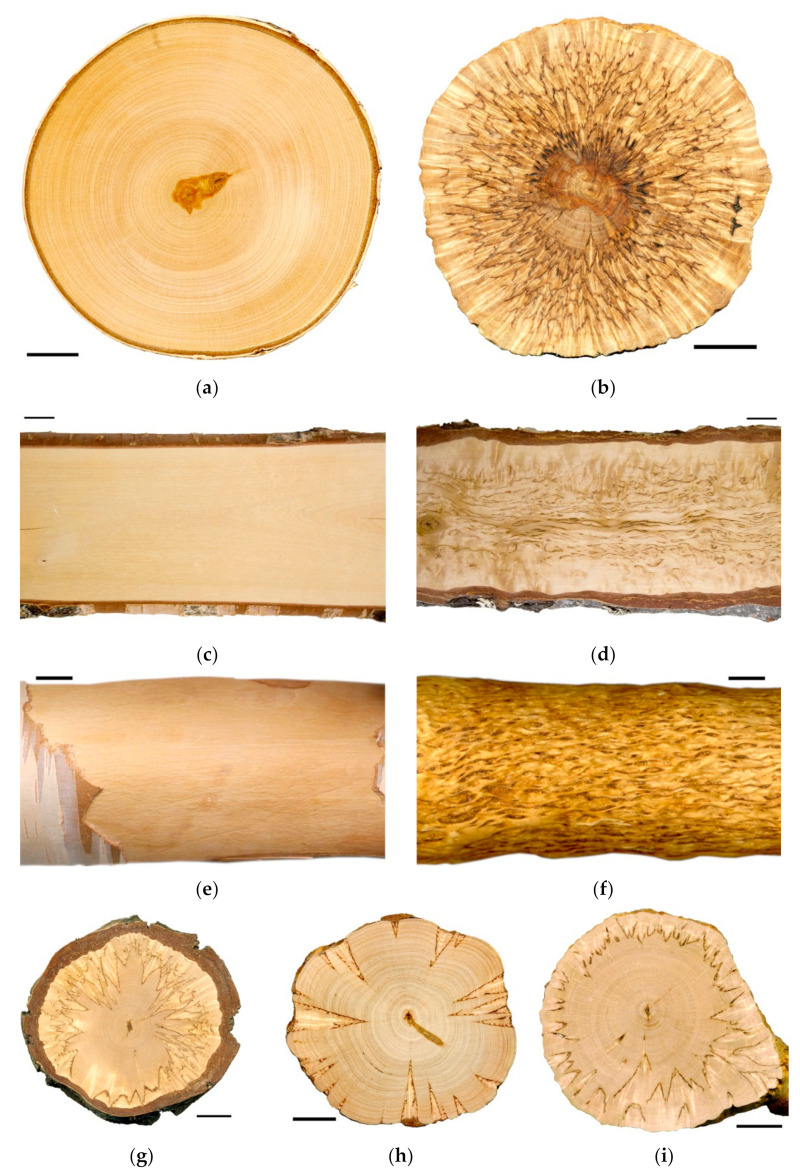
Cross—(**a**,**b**,**g**–**i**) and longitudinal (**c**,**d**) wood sections, debarked wood surface (**e**,**f**) of *B*. *pendula* var. *pendula* (**a**,**c**,**e**) and figured *B*. *pendula* var. *carelica* (**b**,**d**,**f**–**i**) trees. Bar: 2 cm.

**Figure 2 plants-10-01593-f002:**
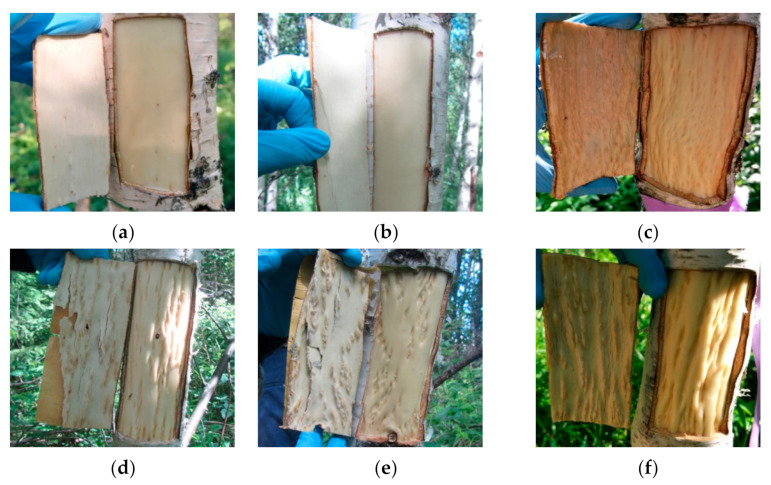
Debarked wood surface and inner bark surface of *B*. *pendula* var. *pendula* (**a**), non-figured (**b**), and figured (**c**–**f**) *B*. *pendula* var. *carelica* trees.

**Figure 3 plants-10-01593-f003:**
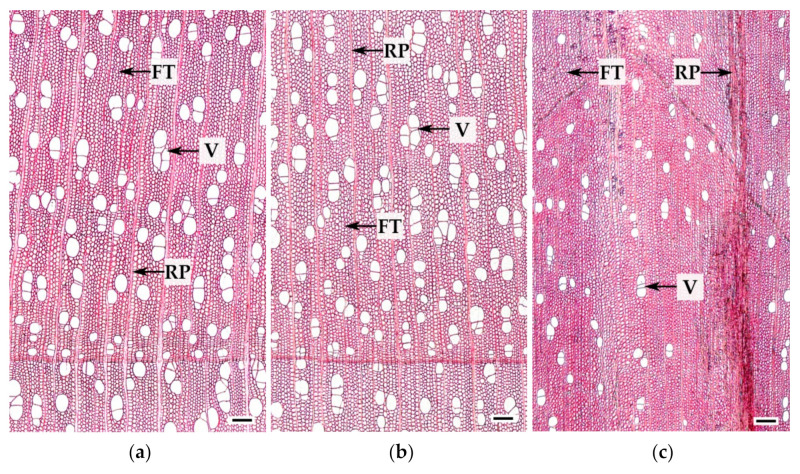
Cross-sections of mature current year xylem of *B. pendula* var. *pendula* (**a**), non-figured (**b**), and figured (**c**) *B. pendula* var. *carelica* trees. V—vessels, FT—fibrous tracheids, RP—radial parenchyma cells. Scale bar: 200 µm.

**Figure 4 plants-10-01593-f004:**
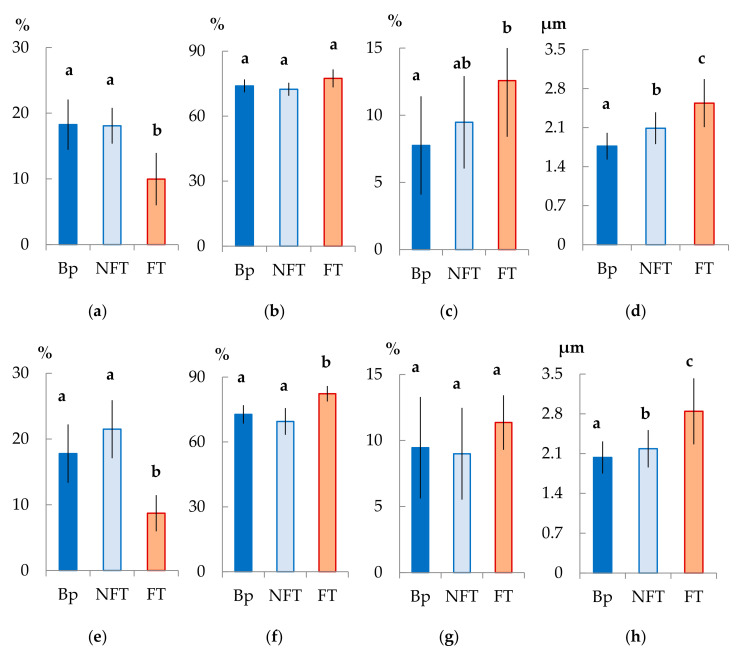
Anatomical parameters of *B. pendula* var. *pendula* (Bp) mature xylem, non-figured (NFT), and figured (FT) *B. pendula* var. *carelica* mature xylem. Samples were collected on 25 June 2019 (**a**–**d**) and 26 June 2020 (**e**–**h**). Proportion (%) of vessels (**a**,**e**), fibrous tracheids (**b**,**f**), radial and axial parenchyma (**c**,**g**), and cell wall thickness (μm) of fibrous tracheids (**d**,**h**) in the mature xylem of the studied trees are shown. Different letters indicate significant differences at *p*-value ˂ 0.05. Bars are means of n experimental runs ± SD according to Mann–Whitney U-test *n* = 9.

**Figure 5 plants-10-01593-f005:**
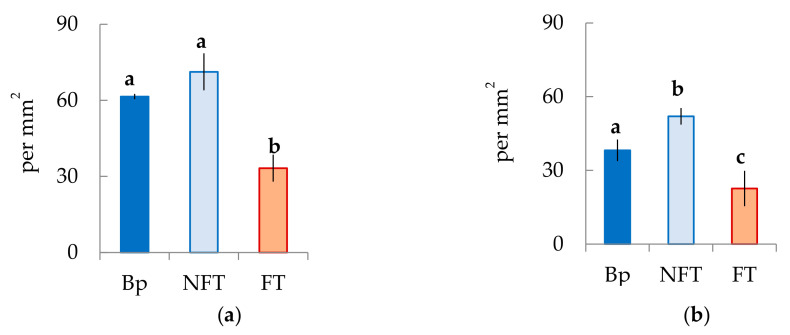
The vessel density is defined as the number of individual vessels per mm^2^ in the mature xylem of *B. pendula* var. *pendula* (Bp), non-figured (NFT), and figured (FT) *B. pendula* var. *carelica* trees. Samples were collected on 25 June 2019 (**a**) and 26 June 2020 (**b**). Different letters indicate significant differences at *p*-value ˂ 0.05. Bars are means of n experimental runs ± SD according to Mann–Whitney U-test. *n* = 9.

**Figure 6 plants-10-01593-f006:**
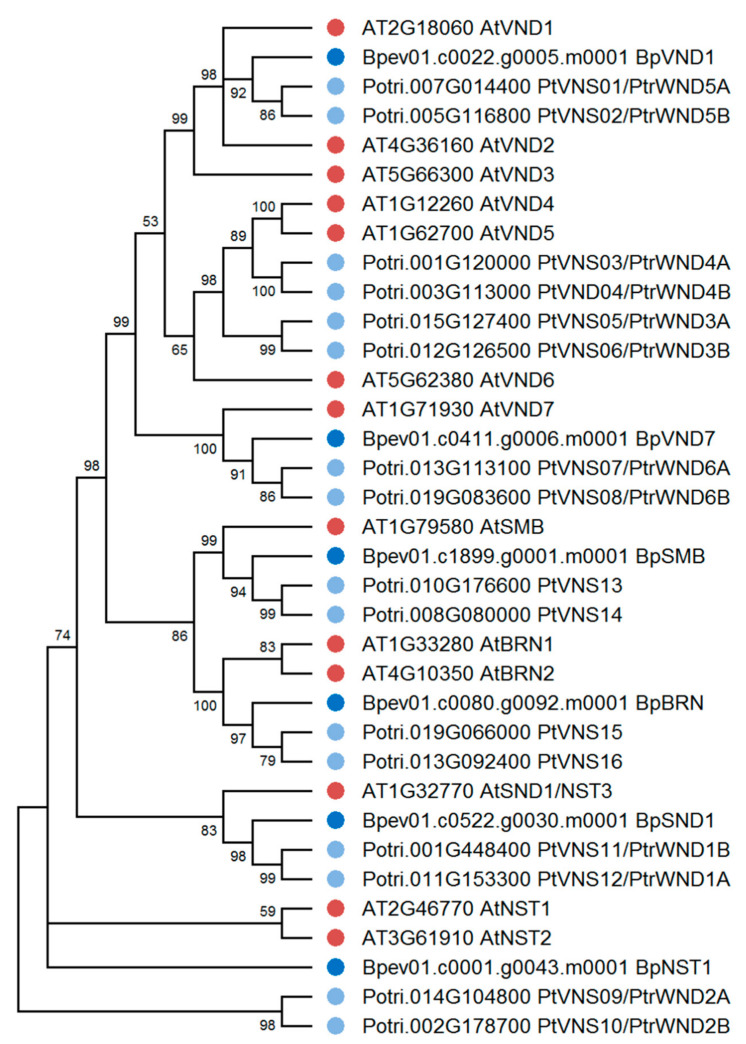
Phylogenetic relationships of Class IIB NACs of *B. pendula* (dark blue dots), *A. thaliana* (red dots), and *P. trichocarpa* (blue dots). Bootstrap values (1000 replicates) are shown next to the branches, and only those values greater than 50 are displayed. The access codes of the *A. thaliana* and *P. trichocarpa* proteins in the TAIR and Phytozome databases are indicated next to the corresponding proteins.

**Figure 7 plants-10-01593-f007:**
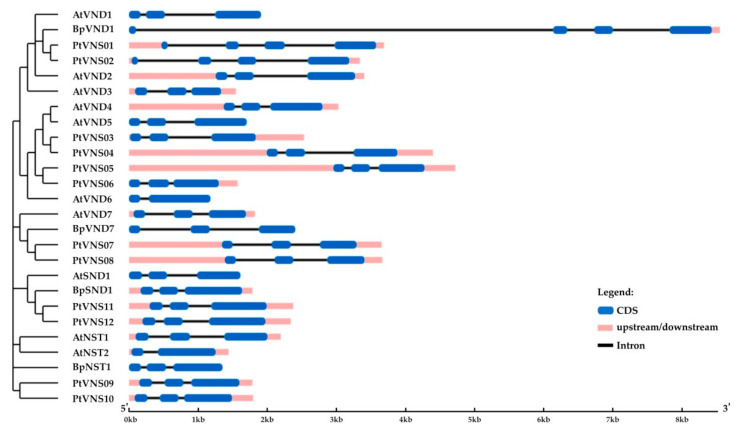
Structure of *B. pendula*, *A. thaliana* and *P. trichocarpa*
*VND* and *NST* genes. Intron, exon, and untranslated regions (UTR) are represented by black lines, dark blue boxes, and pink boxes.

**Figure 8 plants-10-01593-f008:**
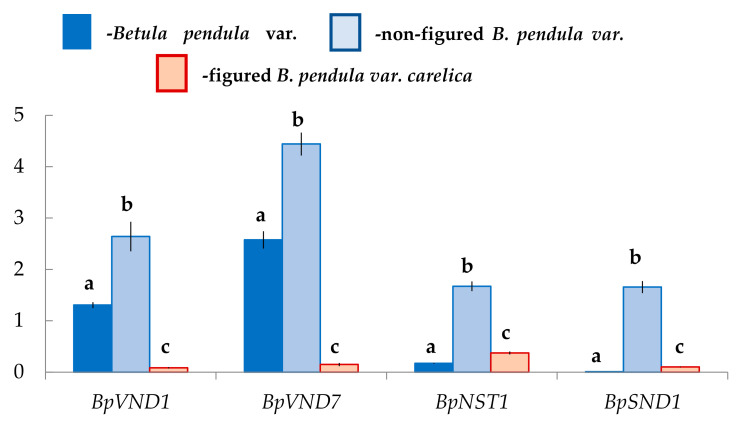
Relative expression (arbitrary units) of genes with the NAC domain in the xylem of *B*. *pendula* var. *pendula*, non-figured and figured *B. pendula* var. *carelica* trees. Samples were collected on 25 June 2019. Different letters indicate significant differences at *p*-value ˂ 0.05. Bars are means of n experimental runs ± SD according to Mann–Whitney U-test. *n* = 6.

**Figure 9 plants-10-01593-f009:**
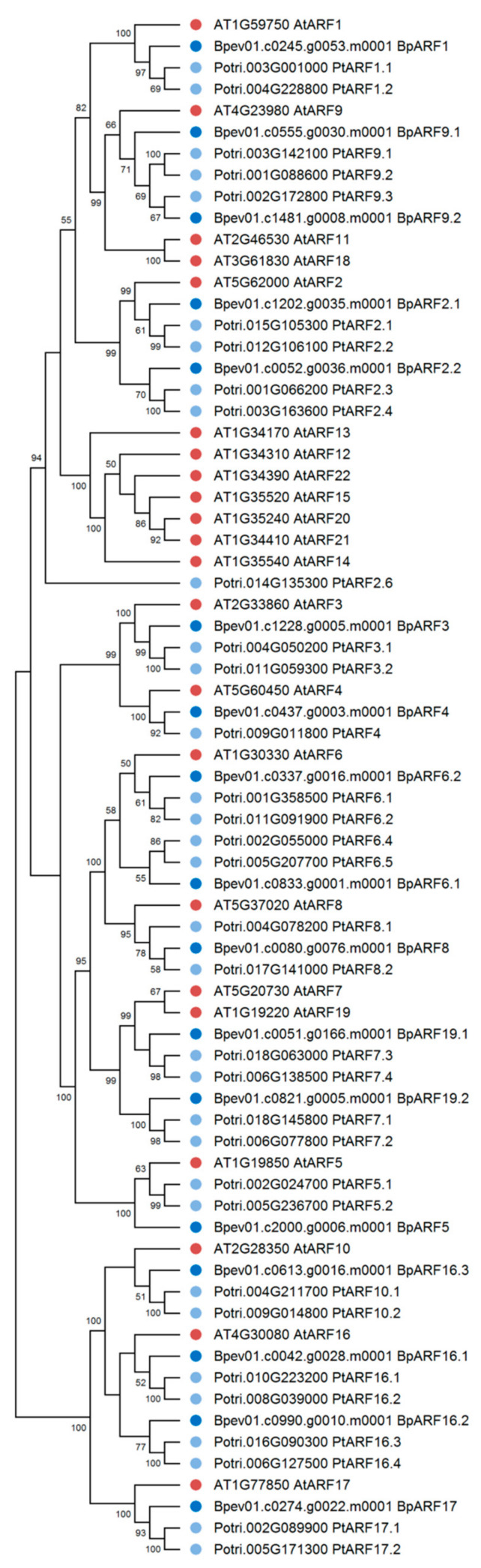
Phylogenetic relationships of ARFs of *B. pendula* (dark blue dots), *A. thaliana* (red dots), and *P. trichocarpa* (blue dots). Bootstrap values (1000 replicates) are shown next to the branches, and only those values greater than 50 are displayed. The access codes of the *A. thaliana* and *P. trichocarpa* proteins in the TAIR and Phytozome databases are indicated next to the corresponding proteins.

**Figure 10 plants-10-01593-f010:**
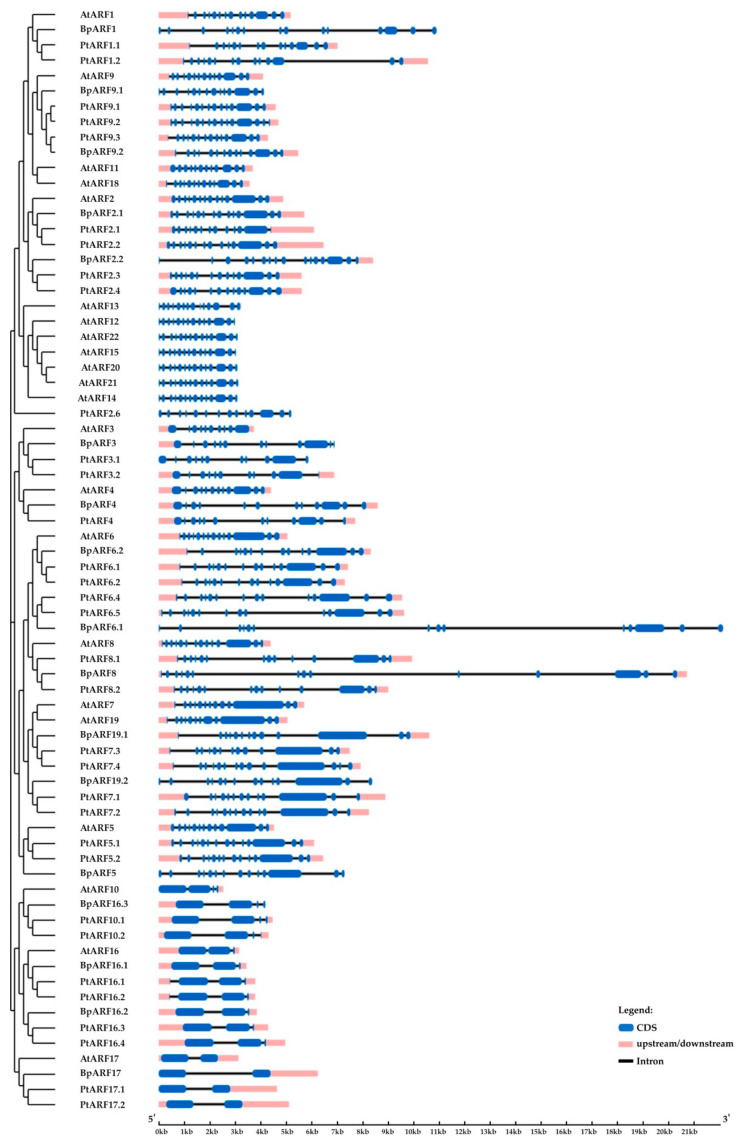
Structure of *B. pendula*, *A. thaliana* and *P. trichocarpa* ARF genes. Intron, exon, and untranslated regions (UTR) are represented by black lines, dark blue boxes, and pink boxes.

**Figure 11 plants-10-01593-f011:**
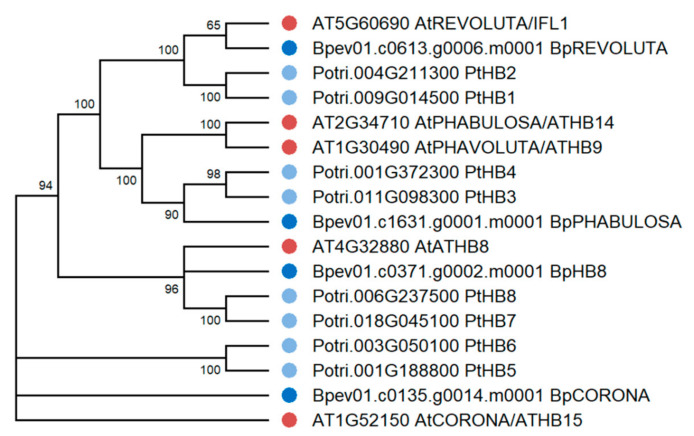
Phylogenetic relationships of HD ZIPs III of *B. pendula* (dark blue dots), *A. thaliana* (red dots), and *P. trichocarpa* (blue dots). Bootstrap values (1000 replicates) are shown next to the branches, and only those values greater than 50 are displayed. The access codes of the *A. thaliana* and *P. trichocarpa* proteins in the TAIR and Phytozome databases are indicated next to the corresponding proteins.

**Figure 12 plants-10-01593-f012:**
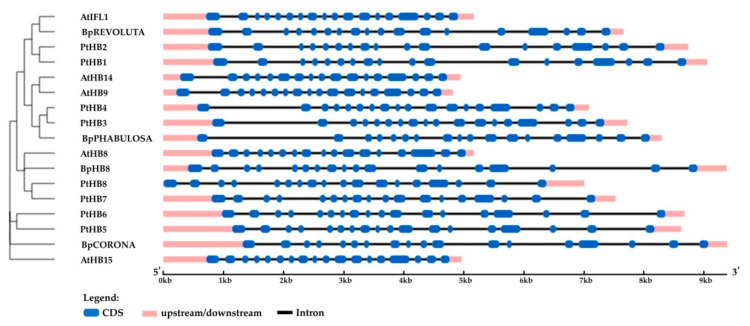
Structure of *B. pendula*, *A. thaliana* and *P. trichocarpa* HD ZIP III genes. Intron, exon, and untranslated regions (UTR) are represented by black lines, dark blue boxes, and pink boxes.

**Figure 13 plants-10-01593-f013:**
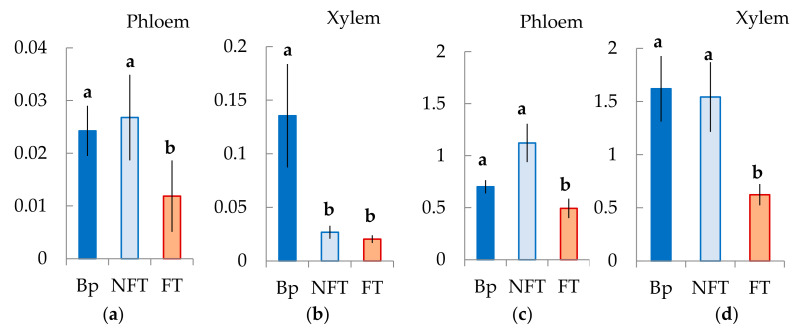
Relative expression (arbitrary units) of genes *BpARF5* (**a**,**b**) and *BpHB8* (**c**,**d**) in tissues including the cambial zone, the differentiating phloem and the mature phloem (Phloem), and the differentiating xylem (Xylem) *B*. *pendula* var. *pendula* (Bp), non-figured (NFT) and figured (FT) *B*. *pendula* var. *carelica* trees. Samples were collected on 26 June 2020. Different letters indicate significant differences at *p*-value ˂ 0.05. Bars are means of n experimental runs ± SD according to Mann–Whitney U-test. *n* = 9.

**Table 1 plants-10-01593-t001:** Characteristics of VND and NST members in birch.

Gene ID	*B. pendula* Protein Name	Protein Length (aa)	Genome Location	Closest Homolog *A. thaliana* (% Identity)	Closest Homolog *P. trichocarpa* (% Identity)
*Bpev01.c0022.g0005*	BpVND1	394	Chr4: 1,788,544–1,796,957	AtVND1 (57.3%)	PtVNS01 (73.5%)
*Bpev01.c0411.g0006*	BpVND7	319	Chr7: 25,758,566–25,760,964	AtVND7 (55.1%)	PtVNS07 (69.2%)
*Bpev01.c0001.g0043*	BpNST1	385	Chr5: 20,598,169–20,599,517	AtNST1 (54.0%)	PtVNS10 (65.9%)
*Bpev01.c0522.g0030*	BpSND1	426	Chr14: 273,787–275,254	AtSND1 (48.9%)	PtVNS11 (69.2%)

**Table 2 plants-10-01593-t002:** Characteristics of ARF members in birch.

Gene ID	*B. pendula* Protein Name	Protein Length (aa)	Genome Location	Closest Homolog *A. thaliana* (% Identity)	Closest Homolog *P. trichocarpa* (% Identity)
*Bpev01.c0042.g0028*	BpARF16.1	688	Chr5: 5,076,067–5,078,769	AtARF16 (59.6%)	PtARF16.2 (67.6%)
*Bpev01.c0051.g0166*	BpARF19.1	1135	Chr10: 19,836,755–19,845,827	AtARF19 (65.2%)	PtARF7.3 (74.8%)
*Bpev01.c0052.g0036*	BpARF2.2	825	Contig52: 293,588–301,375	AtARF2 (47.2%)	PtARF2.3 (48.8%)
*Bpev01.c0080.g0076*	BpARF8	849	Chr2: 10,026,633–10,046,765	AtARF8 (68.2%)	PtARF8.1 (79.6%)
*Bpev01.c0245.g0053*	BpARF1	718	Chr3: 494,058–504,888	AtARF1 (67.3%)	PtARF1.2 (75.7%)
*Bpev01.c0274.g0022*	BpARF17	592	Chr1: 38,790,620–38,794,987	AtARF17 (50.5%)	BpARF17.1 (65.3%)
*Bpev01.c0337.g0016*	BpARF6.2	912	Chr6: 7,510,485–7,517,394	AtARF6 (73.0%)	PtARF6.2 (79.9%)
*Bpev01.c0437.g0003*	BpARF4	799	Chr13: 14,735,340–14,742,867	AtARF4 (62.5%)	PtARF4 (71.2%)
*Bpev01.c0555.g0030*	BpARF9.1	683	Chr14: 1,409,167–1,413,267	AtARF9 (56.9%)	PtARF9.1 (71.6%)
*Bpev01.c0613.g0016*	BpARF16.3	713	Chr13: 13,624,694–13,628,188	AtARF16 (55.3%)	PtARF16.2 (61.1%)
*Bpev01.c0821.g0005*	BpARF19.2	1109	Chr7: 4,843,289–4,851,608	AtARF19 (48.9%)	PtARF7.1 (79.8%)
*Bpev01.c0833.g0001*	BpARF6.1	894	Chr2: 16,657,839–16,679,834	AtARF6 (67.6%)	PtARF6.5 (80.8%)
*Bpev01.c0990.g0010*	BpARF16.2	702	Chr8: 5,039,586–5,042,474	AtARF16 (58.6%)	PtARF16.4 (76.2%)
*Bpev01.c1202.g0035*	BpARF2.1	841	Chr12: 25,724,880–25,729,200	AtARF2 (66.6%)	PtARF2.2 (75.3%)
*Bpev01.c1228.g0005*	BpARF3	753	Chr6: 23,740,690–23,746,958	AtARF3 (47.5%)	PtARF3.1 (64.1%)
*Bpev01.c1481.g0008*	BpARF9.2	690	Chr5: 21,299,075–21,303,311	AtARF9 (55.1%)	PtARF9.1 (68.7%)
*Bpev01.c2000.g0006*	BpAFR5	933	Chr9: 2,615,628–2,622,870	AtARF5 (58.9%)	PtARF5.2 (70.5%)

**Table 3 plants-10-01593-t003:** Characteristics of HD ZIP III members in birch.

Gene ID	*B. pendula* Protein Name	Protein Length (aa)	Genome Location	Closest Homolog *A. thaliana* (% Identity)	Closest Homolog *P. trichocarpa* (% Identity)
*Bpev01.c0135.g0014*	BpCORONA	836	Chr5: 19,578,340–19,586,093	AtCORONA/ATHB15 (88.3%)	PtHB6 (92.7%)
*Bpev01.c0371.g0002*	BpHB8	837	Chr9: 13,192,071–13,200,917	AtHB8 (80.8%)	PtHB8 (90.0%)
*Bpev01.c0613.g0006*	BpREVOLUTA	843	Chr13: 13,883,876–13,890,572	AtREVOLUTA/IFL1 (84.6%)	PtHB1 (86.3%)
*Bpev01.c1631.g0001*	BpPHABULOSA	833	Chr6: 9,307,551–9,315,088	AtPHABULOSA/ATHB14 (80.7%)	PtHB4 (91.7%)

**Table 4 plants-10-01593-t004:** Presence of known and putative AuxRE *cis*-elements in promoters of studied *B. pendula* genes. Numbers indicate the distance from the start of transcription, plus or minus in brackets shows strand.

Gene Name	Known AuxREs		Putative AuxREs	
	TGTCTC	TGTCCC	TGTGGG	GTCCCC
*BpHB8*		1273 (−), 471 (+)		470 (+)
*BpVND1*	458 (−), 46 (+)		1666 (−), 447 (+)	
*BpVND7*	805 (−), 703 (−), 28 (+)	1956 (+)	1427 (+), 372 (−), 308 (−)	
*BpNST1*	908 (−), 601 (−)	1554 (−)	491 (−)	
*BpSND1*	1630 (−)		1746 (+), 288 (−)	709 (+)

**Table 5 plants-10-01593-t005:** List of primers for the RT-PCR reaction.

Gene Name	Gene ID	Forward Primer (5′→3′)	Reverse Primer (5′→3′)	Tm, °C
*Ef1a*	Bpev01.c0437.g0013	TCCTTGAGGCTCTTGACTTG	ATACCAGGCTTGATGACACC	89.0
*GAPDH*	Bpev01.c1040.g0016	AGAATACAAGCCAGAACTCAAC	CTCTACCACCTCTCCAATCC	85.9
*BpARF5*	Bpev01.c2000.g0006	AGCGACACCTTCTCACAACTG	GCCTCACACCAACCAATAACTG	84.9
*BpHB8*	Bpev01.c0371.g0002	GTAGTGGAGTGGATGAGAATG	TCAAGAGCAGAGGCAAGG	88.0
*BpVND1*	Bpev01.c0022.g0005	AAGAGATAGAAGCGGATA	TTATTAACGGCAGAGATG	76.8
*BpVND7*	Bpev01.c0411.g0006	TTATGAACAGAATGAGTGGTA	GCAGTGGCTCTATTAGTC	74.3
*BpNST1*	Bpev01.c0001.g0043	AGTATATCAGTCAATGGA	AGTATATCAGTCAATGGA	77.2
*BpSND1*	Bpev01.c0522.g0030	AACTTCTACACTACTACC	CACATCTCTTGAATATCC	76.8

## Data Availability

Not applicable.

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
