# Peer review of "Molecular Genetic Characteristics of Different Scenarios of Xylogenesis on the Example of Two Forms of Silver Birch Differing in the Ratio of Structural Elements in the Xylem"

_plants, 2021, doi:10.3390/plants10081593_

Round 1
Reviewer 1 Report
The new version of the manuscript has been significantly improved. The authors referred to my comments by introducing corrections or removing parts of the text. However, it still could use a better justification for the research. Why were these two forms of birch selected for this study? What was expected? Research hypotheses should be better exposed. The authors of taxon names should be checked and corrected according to International Plant Names Index or The Plant Names databases.
Author Response
The two forms of silver birch differ significantly in the structural elements ratio in the mature xylem. The way of differentiation of cambial derivatives into vessels and fibrous tracheids predominates in silver birch, and in Karelian birch the cambial derivatives differentiate into parenchymal cells in places of anomalies. We examined the assignment of expression genes, encoding NAC-domain transcription factors that regulated secondary cell wall synthesis, development and differentiation of xylem cells; transcription factor BpHB8 that promoted xylem production from the cambial cells, as well as auxin-dependent transcription factor BpARF5 under different xylogenesis scenarios. We have identified for the first time BpARF5, which was a key regulator of auxin-dependent vascular pattern, and its direct target, BpHB8, and showed a change in their gene expression level depending on the tree phenotype.
We checked and corrected the taxon names according to International Plant Names Index.
Thanks.
Reviewer 2 Report
The manuscript by (Galibina et al) focused on the molecular background of different xylogenesis scenarios in the birch trunks. They identified the genes that regulate secondary cell wall biosynthesis in the birch. Most importantly they identified BpARF5, which is a key regulator of auxin-dependent vascular patterning, and its direct target - BpHB8. The content of the manuscript is clear concise and simply presented. The figures are descriptive and merging with the theme of the text. There are not many mistakes in the text. In brief, after possible comments are addressed by the authors, I would support this manuscript based on the content and the importance of the field for acceptance in this journal.
My comment is as follows:
Line 252- Authors have done the cross-section images to visualize the density and the cell wall thickness of fibrous tracheids and vessels. Did the authors check the length of these after performing maceration on wood tissue samples for e.g, into individual vessels? This might give more information on their anatomy. Authors should also have checked the cell wall thickness after maceration. Can the authors provide an explanation for that?
Minor spelling mistakes:
For e.g, in Line 242- thickness
Author Response
We investigated the anatomical and morphological features of individual structural elements of wood (fibrous tracheids and vessels) in silver and Karelian birch with the help of the macerated material. Our previous studies showed that the length of fibrous tracheids and vessels was shorter in Karelian birch, and the width of vessel was shorter. We are planning research in this field.
We rechecked the entire text of the manuscript.
Thanks.
Reviewer 3 Report
I carefully read the manuscript titled "Molecular Genetic Characteristics of Different Scenarios of Xylogenesis (on the Example of Two Forms of Silver Birch Differing in the Ratio of Structural Elements in the Xylem)", and I see it was revised already after previous reviewers comments.
The manuscript is in many parts hard to read, therefore I would suggest a revision to improve the writing towards clarity before its final publication.
I have only a few specific comments which I list below:
L49 chance "parenchymal" for "parenchyma"
L61 please specify what is meant by "formation of both softwood and hardwood". Wood formation?
in Fig. 1 Caption replace "cross-sections" with surfaces. Transverse and cross-section are synonyms.
L628 the (beautiful!) anatomical cross-sections shown in fig 3 looks thicker than 2 micrometres. Was the section thickness really 2 microns or actually 20?.
I wonder if figure 2 can be somehow incorporated into fig 1
Author Response
We rechecked the entire text of the manuscript and and made the changes suggested by you.
The section thickness is really 2. There is no mistake here.
Figures 1 and 2 show photographs of different objects, so we do not consider it logical to combine these figures. Figures 1 represents how both forms of birch can look outwardly. Figure 2 shows live photos of the objects of study.
We rechecked the entire text of the manuscript.
Thanks.